# Parental Awareness, Knowledge, and Attitudes Regarding Current and Future Newborn Bloodspot Screening: The First Report from Thailand

**DOI:** 10.3390/ijns9020025

**Published:** 2023-05-03

**Authors:** Kalyarat Wilaiwongsathien, Duangrurdee Wattanasirichaigoon, Sasivimol Rattanasiri, Chanatpon Aonnuam, Chayada Tangshewinsirikul, Thipwimol Tim-Aroon

**Affiliations:** 1Department of Pediatrics, Faculty of Medicine Ramathibodi Hospital, Mahidol University, Bangkok 10400, Thailand; k.wilaiwongsathien@hotmail.com; 2Division of Medical Genetics, Department of Pediatrics, Faculty of Medicine Ramathibodi Hospital, Mahidol University, Bangkok 10400, Thailand; duangrurdee.wat@mahidol.ac.th; 3Department of Clinical Epidemiology and Biostatistics, Faculty of Medicine Ramathibodi Hospital, Mahidol University, Bangkok 10400, Thailand; 4Department of Obstetrics and Gynecology, Faculty of Medicine Ramathibodi Hospital, Mahidol University, Bangkok 10400, Thailand

**Keywords:** newborn screening, bloodspots, awareness, knowledge, attitude, parental perspectives, Thailand, Southeast Asia

## Abstract

Newborn screening (NBS) is a public health service that is used to screen for treatable conditions in many countries, including Thailand. Several reports have revealed low levels of parental awareness and knowledge about NBS. Because of limited data on parental perspectives toward NBS in Asia and the differences in socio-cultural and economic contexts between Western and Asian countries, we conducted a study to explore parental perspectives on NBS in Thailand. A Thai questionnaire to assess awareness, knowledge, and attitudes regarding NBS was constructed. The final questionnaire was distributed to pregnant women, with or without their spouses, and to parents of children aged up to one year who visited the study sites in 2022. A total of 717 participants were enrolled. Up to 60% of parents were identified as having good awareness, which was significantly associated with gender, age, and occupation. Only 10% of parents were classified as having good knowledge relative to their education level and occupation. Providing appropriate NBS education should be initiated during antenatal care, focusing on both parents. This study noted a positive attitude toward expanded NBS for treatable inborn metabolic diseases, incurable disorders, and adult-onset diseases. However, modernized NBS should be holistically evaluated by multiple stakeholders in each country because of different socio-cultural and economic contexts.

## 1. Introduction

Newborn screening (NBS) programs are supported by national health policies to prevent serious health problems in most countries. Simple blood spots are collected shortly after birth, before symptoms present themselves. All diseases screened in the NBS are treatable conditions, and treatment usually starts in the neonatal or infancy period to reduce and prevent morbidity and mortality. The conditions screened differ from country to country or even between different states in the same country [1,2].

Although NBS has been practiced for over 60 years in many countries, several reports have revealed low parental awareness and knowledge about NBS [3,4,5,6,7,8,9]. It has been observed that factors associated with the levels of awareness and knowledge include the age of the mothers, the number of children, and parental income and educational levels [5,8,10,11,12]. Increasing parental awareness and knowledge levels through adequate education would aid in improving neonatal health care. In Thailand, NBS for congenital hypothyroidism (CH) and phenylketonuria (PKU) has been provided since 1996 [13,14]. A Thai NBS program is a mandated program that is covered under universal coverage for all Thai newborns. There is no parental written consent for NBS. Consent is required to share the leftover blood spots for future research purposes. Since 2002, the NBS coverage rate for CH and PKU in Thailand has been over 85%, and more than 95% for the last 10 years [15]. Expanded NBS for inborn metabolic diseases using MSMS has been part of the Thai universal coverage program as a mandatory NBS since October 2022, but it was launched after this present study.

Because of the limited data on parental perspectives toward NBS in Asia and the differences in socio-cultural contexts between Western and Asian countries, we conducted this study to assess Thai parents’ awareness of and knowledge and attitudes about NBS. Moreover, we investigated NBS education and parental attitudes toward future trends in expanded NBS.

## 2. Materials and Methods

### 2.1. Study Design and Population

A cross-sectional study was conducted from June 2021 to October 2022. Pregnant women with or without their spouses and parents of children aged 0–1 years who visited the antenatal care clinic, postpartum unit, pediatric outpatient clinic, or inpatient wards at Ramathibodi Hospital and Maharat Nakhon Ratchasima Hospital were invited to participate in the study. Ramathibodi Hospital is a university hospital in Bangkok, and Maharat Nakhon Ratchasima Hospital is a tertiary hospital in Northeastern Thailand. Both hospitals are considered to be large maternity hospitals. The eligible participants were aged ≥15 years and could read Thai. Pregnant women admitted for pregnancy termination were excluded to avoid psychological stress. 

All study participants provided written informed consent before data collection. Pen-and-paper questionnaires were distributed to the participants. The questionnaires took approximately 10–15 min for self-completion. The researcher was nearby if the participants had any questions about the questionnaires. 

The study protocol was approved by the Human Research Ethics Committee, Faculty of Medicine Ramathibodi Hospital, Mahidol University, and Maharat Nakhon Ratchasima Hospital Institutional Review Board (MURA2021/413 and 047/2022).

### 2.2. Questionnaire

A Thai questionnaire for NBS was developed based on information gathered from literature reviews and expert consultation. The questionnaire consisted of questions concerning four areas: awareness, NBS education, knowledge, and attitudes about current and future NBS. Using the index of item objective congruence (IOC), questionnaire validity was assessed by four clinical experts: two neonatologists, a pediatric geneticist, and an obstetrician. The evaluation of questionnaire clarity was performed on a small group of candidate participants at Ramathibodi Hospital with a Cronbach’s alpha coefficient of 0.79.

The final version of the questionnaire consisted of 4 parts, including 31 questions. The part concerning parental awareness (4 questions) consisted of know/don’t know/uncertain questions about 3 general aspects of NBS, including the existence, safety, and importance of NBS, and overall self-reported NBS awareness. We defined participants as having good awareness if the individuals knew about all three items. NBS education (3 questions) consisted of multiple choice questions about NBS resources and short answers on the optimal time for NBS education. 

The evaluation of parental knowledge about NBS (12 questions) consisted of NBS facts, including screened diseases, the purpose of NBS, the blood collection process, and clinical consequences if not treated. These consisted of true/false questions. A score of 1 was given to correct answers, and a score of 0 was given to incorrect or uncertain answers. The total scores were calculated for each participant. The maximum score was 12/12. A participant with a total score of ≥60% (7 of 12 items) was classified as having good knowledge.

For the attitude assessment (12 questions), participants were asked to complete a 12-item questionnaire on a 5-point Likert-type scale. The answers were classified into three groups: agree (strongly agree or agree), uncertain, and disagree (strongly disagree or disagree). The questions covered current NBS topics consisting of parental concerns about false-positive results, informed consent, and research using leftover dry blood, as well as future NBS issues including treatable inborn metabolic diseases, incurable disorders, and adult-onset diseases. Moreover, the questionnaire included short answer questions about why the participants agreed or disagreed with screening for incurable or adult-onset diseases. We also asked their opinions on the reasonable cost of future expanded NBS if the participant was willing to pay.

### 2.3. Statistical Analysis

Categorical variables were analyzed using simple descriptive statistics. Univariate and multivariate logistic regression analyses were performed to identify factors affecting awareness and knowledge. The dependent variables used a recode outcome (strongly agree or agree/strongly disagree or disagree/neither agree nor disagree). All statistical tests were significant level *p* ≤ 0.05 as analyzed by STATA version 17.

## 3. Results

The questionnaire was administered to 750 eligible parents. A total of 729 parents participated in the study. Twelve participants were excluded as they presented significantly incomplete answers. Questionnaires from 717 participants were analyzed, representing a response rate of 95.6%. The participant characteristics are shown in Table 1. Most participants were female (83.3%), and the age of participants ranged from 15 to 57 years, with a median age of 32 years. Most of the participants had one child or more (69.7%), did not have a career in healthcare (90%), had a monthly income below THB 25,000 or USD 730 (USD 1 approximately equal to THB 34.5) (71.1%), and had not previously had a child with a genetic disease (97.1%). Nearly half of the participants had education below a bachelor’s degree (47.3%). The participants were from all six regions of Thailand. Approximately 80% of participants came from the central and northeastern parts of Thailand, which have the largest populations in the country. 

### 3.1. Parental Awareness

More than two-thirds of participants reported being aware of the existence of the NBS in Thailand for all neonates (70.1%), the safety of the NBS (77%), and the importance of taking immediate action in the event of a positive NBS (78%). Nearly half of the participants’ self-rated scores for awareness of NBS were uncertain (score 5 out of 10). 

Sixty percent of participants were identified as having good awareness. Univariate and multivariate analysis demonstrated that having good awareness was significantly associated with female gender (OR 2.02; 95%CI 1.34–3.03), age of 21–34 years (OR 1.40; 95%CI 1.00–1.95), and being a healthcare provider (OR 2.43; 95%CI 1.33–4.43) (Table 2). 

### 3.2. NBS Education

About one-third (32.4%) of participants reported never receiving information about NBS. The 470 participants who reported receiving NBS information recalled that they were informed by a healthcare provider (66.3%), family and friends (19.9%), social media (14.4%), and brochures (9%), whereas one-fourth (25.7%) reported their own experience from having a previous child. When asked about the optimal time for NBS education, two-thirds of parents (66.3%) suggested NBS education should be provided during the antenatal period (first trimester 43.1%, second trimester 11.5%, third trimester 11.7%), public education 25.7%, and the postpartum period 8%.

### 3.3. Parental Knowledge 

The results for parental knowledge about NBS are shown in Table 3. Only 41% of the participants identified CH, and 22.3% correctly identified PKU as the screened disease. A small percentage of the participants correctly indicated that diabetes mellitus (DM) (6.7%), asthma (10.4%), and cancer (14.1%) cannot screen by NBS. Most participants knew two main facts, including the aims of NBS (79.9%) and the parental action needed in the event of positive NBS (91%). Nearly 75% of the participants need more knowledge about the necessity of a confirmatory test for positive NBS.

Interestingly, only 10% of participants were classified as having good knowledge (score ≥ 7). Statistical analysis demonstrated that having good knowledge is associated with a higher educational level (OR 2.56; 95%CI 1.14–5.75) and with healthcare providers (OR 3.57; 95%CI 1.94–6.56) (Table 4).

### 3.4. Parental Attitudes toward NBS

Parental attitudes about the current and future NBS programs are presented in Table 5. Most participants would prefer to be informed by healthcare providers (94.2%) and to sign a consent form before blood collection (91.1%). The participants accepted the possibility of false-positive results from NBS (70.7%). Only 60% of the participants agreed that the leftover blood spots from their babies could be shared for future research without personal identification. Interestingly, 25% of the participants agreed that the screened diseases should include only treatable conditions. About 57% of the parents agreed that they are anxious and that it may impact their childcare if their babies have false-positive results. 

Regarding an expanded NBS for treatable inborn metabolic diseases, more than 80% of the participants supported the national coverage program. Only 65% of participants are willing to pay for expanded NBS if needed, and 74.2% would prefer to pay less than THB 1000 (USD 30).

Surprisingly, more than 80% of participants supported future newborn screening for incurable and adult-onset disorders, if made available for the reasons of child-care preparation (91%), family planning (50%), and research and new treatment opportunities (50%). Less than 5% of the participants disagreed with screening for incurable or adult-onset diseases, the reasons for which included ongoing anxiety (66.7%), the difficulty of living with uncertainty (24%), and issues with obtaining insurance (14%).

## 4. Discussion

This is the first cross-sectional survey that evaluates parental awareness and education, knowledge, and attitudes regarding the current NBS and future expanded NBS in Thailand. Parental awareness was shown to be limited: only 59.7% of parents were classified as having good awareness, and our study found that females had good awareness, of about two times more than males. The finding is similar to that of a previous study conducted in Japan in 2010, which showed low awareness of NBS (26.6%), with a lower rate in males (13.2%) than in females (40.4%) [3]. It is essential to focus on both parents to increase awareness. Encouraging the father’s involvement in NBS will benefit the child’s health. Our study found that parents with an age range of 21–34 years had better awareness (1.4 times higher) than the other age groups. Meanwhile, a study in the Czech Republic in 2019 showed good awareness in older age groups (≥30 years) and among parents with multiple children [10]. These differences might be because, nowadays, young parents easily assess and seek out information from public resources. Interestingly, education level and income are not related to the level of awareness in the present study. However, research from the Philippines in 2015 found that the parents’ education level and monthly income were significantly associated with good awareness [12]. These differences might be because the two studies use different categorizations of education level and income. 

In the present study, one-third of parents reported never receiving information about NBS. In this study, the parents stated that the most appropriate time for NBS education is during the antenatal period, especially during the first trimester. In previous studies, the participants suggested that the antenatal period, especially in the third trimester, is the ideal time to introduce NBS knowledge, because they are approaching the time when they will become mothers and are preparing themselves for the baby [5,10,16]. The postpartum period is not a good time to introduce NBS education because mothers are physically exhausted and overwhelmed with a large amount of information about newborn care [10,16]. This study therefore advocates informing both parents about NBS during antenatal care, changing the current approach. Our study also found that one-fourth of parents thought NBS should be a part of local public education. 

Nearly half of the parents reported in their self-assessment that they were uncertain about their awareness of NBS; however, only 10% of parents were classified as having good knowledge. This study showed a significant risk of misunderstanding, especially regarding the currently screened diseases. Approximately 40% of parents correctly identified CH, and about 22% correctly identified PKU as the screened disease in this study. The finding is consistent with a study conducted in Ireland, which found that pregnant women identified CH and PKU at rates of 13% and 26%, respectively [5,6]. Surprisingly, more than 85% of the parents misunderstood or were unsure whether common diseases such as DM and asthma were included in the current NBS. These values are significantly higher than those found in previous studies, which indicated that women incorrectly identified common conditions tested for NBS in Ireland and Saudi Arabia at rates of 20% and >50%, respectively [5,6,9]. This finding indicates a need for detailed education about NBS.

Our study identified that both an educational level higher than a bachelor’s degree and healthcare providers were associated with good knowledge. These factors are similar to those identified in a previous study in Canada [11]. However, the number of children was not found to be related to the level of knowledge in this study, which differs from a previous report. A publication in Ireland reported primiparity (OR 2.75) to be associated with poor NBS knowledge [5].

A positive attitude toward NBS was observed in this study. Most parents accepted the possibility of false-positive results (70%); however, 60% of participants agreed that they experienced ongoing anxiety after false-positive NBS results. Several studies reveal parental anxiety and long-term stress after a false-positive result from NBS, even for treatable conditions such as SCID [17,18]. Some parents additionally experienced parenting problems. Clear information from the healthcare provider both prior to the NBS program and after an abnormal screening result is of the utmost importance for parents [17,18]. 

About 60% of participants in this study agreed with sharing their baby’s blood spots for future research without personal identification, which is similar to the result presented in previous publications [5,9]. However, 15% of participants disagreed with sharing the blood spots for future research. Therefore, consent for leftover bloodspot usage should be clearly informed, and parents have the right to make a decision about research usages. 

Furthermore, in several studies, most parents agreed with pre-test informed consent before blood collection [19,20,21]. Appropriate parental NBS consent is still an issue in several countries [10,19,20,21]. In some countries, only verbal consent is required for NBS; elsewhere, written consent is required for opting out only. No written consent for opting in for NBS has been practiced for many years in Thailand, but consent for sharing leftover specimen for research is provided. Although our study found that most parents prefer pre-test counseling and obtain explicit parental consent for NBS. Because limited parental knowledge of NBS was observed in this study, it might not be appropriate for parental consent to determine whether to opt in or opt out. Parental autonomy in the context of adequate information and public education is necessary.

The parents also indicated their interest in receiving information from newborn genomic sequencing [22,23]. Genomic sequencing would have the added complexity of positive/negative/inconclusive results, which requires more time for parental education.

For future NBS in the genomic era, it would technically be possible to screen for incurable diseases and adult-onset diseases. Although the present study and previous studies revealed parental preferences for the screening of incurable and adult-onset diseases, with sensible reasons given [16,20], psychosocial and ethical issues must be considered. The given reasons included parents’ need to prepare themselves to cope with unfortunate complications/events, to seek new treatment opportunities for their child, to eliminate the need for the diagnostic ordeal that might take place after birth, and the psychological harm that results from not knowing what is wrong with their child, as well as the desire to plan for their child’s future and their own reproductive choices. A previous publication revealed parental stress, anxiety, and depression after positive screening and diagnosis with variable-onset disorders [24]. Therefore, psychosocial and ethical issues have to be considered and require appropriate guidelines or expert opinions based on holistic perspectives and future research. 

The National Health Security Office (NHSO) in Thailand launched universal healthcare coverage for an expanded NBS for 40 treatable inborn metabolic diseases in October 2022, a few months after this questionnaire study was completed. This study will help Thailand and countries with the same socio-economic and cultural contexts to understand parental perspectives and provide a suitable basis for parental education. The modernized NBS needs to be evidence-based and carefully and holistically evaluated by multiple stakeholders in each country [25].

## 5. Conclusions

This is the first study on parental perspectives on NBS in Southeast Asia. It will help us to understand the current status of parental awareness, NBS education, and knowledge and attitudes regarding NBS in Thailand and Southeast Asia contexts. Appropriate information should be provided to both parents during antenatal care. Furthermore, healthcare providers and all stakeholders involved in the NBS program should be able to understand modernized NBS because a worldwide trend of positive parental attitudes toward expanded NBS has been observed, even in developing countries. 

## Figures and Tables

**Table 1 IJNS-09-00025-t001:** Demographic characteristics of participants (n = 717).

Variable	n (%)
**Gender**	
Female	597 (83.3)
Male	120 (16.7)
**Age (years)**	
≤20	44 (6.1)
21–34	439 (61.2)
≥35	234 (32.7)
**Number of children (n = 703)**	
0	213 (30.3)
1	270 (38.4)
≥2	220 (31.3)
**Research site (n = 717)**	
Ramathibodi Hospital	550 (76.7)
Maharat Nakhon Ratchasima Hospital	167 (23.3)
**Hometown (n = 712)**	
Northern	33 (4.6)
Northeastern	300 (42.1)
Central	292 (41.0)
Eastern	26 (3.7)
Western and Southern	61 (8.6)
**Place (n = 692)**	
Antenatal care clinic	375 (54.2)
Postpartum ward	226 (32.7)
Pediatric OPD and IPD	91 (13.1)
**Education level (n = 701)**	
Below bachelor’s degree	332 (47.3)
Bachelor’s degree	311 (44.4)
Master’s degree or higher	58 (8.3)
**Occupation (n = 711)**	
Healthcare provider	71 (10)
Non-healthcare provider	640 (90)
**Income (THB per month) (n = 690)**	
<11,000	172 (24.9)
11,000–24,999	319 (46.2)
≥25,000	199 (28.9)
**Previous child with genetic disease (n = 716)**	
Yes	21 (2.9)
No	695 (97.1)

**Table 2 IJNS-09-00025-t002:** Comparison of participant characteristics between participants with good awareness and those with poor awareness.

	Awareness		Univariate Analysis		Multivariate Analysis	
Participant Characteristics	Good, n (%)	Poor, n (%)	OR (95% CI)	*p*-Value	OR (95% CI)	*p*-Value
	n = 416	n = 281				
**Gender**						
Female	365 (62.9)	215 (37.1)	2.20 (1.47–3.29)	<0.001	2.02 (1.34–3.03)	0.001 *
Male	51 (43.6)	66 (56.4)	1		1	
**Age, years**						
≤20	24 (58.5)	17 (41.5)	1.25 (0.64–2.45)	0.516	1.20 (0.61–2.37)	0.603
21–34	270 (63.4)	156 (36.6)	1.53 (1.11–2.12)	0.010	1.40 (1.00–1.95)	0.049 *
≥35	122 (53)	108 (47)	1		1	
**Number of children**						
Pregnancy with first child	126 (61.2)	80 (38.8)	1.08 (0.77–1.50)	0.668		
Parents of one child or more	284 (59.4)	194 (40.6)	1			
**Place**						
Antenatal care clinic	221 (60.7)	143 (39.3)	1.12 (0.70–1.80)	0.635		
Postpartum ward	133 (60.5)	87 (39.5)	1.11 (0.67–1.83)	0.686		
Pediatric OPD and IPD	51 (58)	37 (42)	1			
**Education level**						
Below bachelor’s degree	204 (63)	120 (37)	1.89 (1.07–3.33)	0.028		
Bachelor’s degree	175 (57.8)	128 (42.2)	1.52 (0.86–2.68)	0.149		
Master’s degree or higher	27 (47.4)	30 (52.6)	1			
**Occupation**						
Healthcare provider	54 (78.3)	15 (21.7)	2.67 (1.48–4.83)	0.001	2.43 (1.33–4.43)	0.004 *
Non-healthcare provider	358 (57.4)	266 (42.6)	1		1	
**Income (THB per month)**						
<11,000	60 (63.2)	35 (36.8)	1.25 (0.75–2.07)	0.388		
11,000–24,999	185 (59.7)	125 (40.3)	1.08 (0.75–1.55)	0.686		
≥25,000	114 (57.9)	83 (42.1)	1			
**Previous child with a genetic disease**						
Yes	10 (47.6)	11 (52.4)	0.60 (0.25–1.44)	0.257		
No	406 (60.1)	270 (39.9)	1			

*significant difference *p*-value < 0.05

**Table 3 IJNS-09-00025-t003:** Number of correct/incorrect/not sure answers to questions evaluating NBS knowledge.

Questions/Statements	Correct Answer	Not Sure	Incorrect Answer
	n (%)	n (%)	n (%)
What conditions are included in the current NBS in Thailand?			
1. Congenital hypothyroidism (n = 688) (true)	282 (41)	388 (56.4)	18 (2.6)
2. PKU (n = 672) (true)	150 (22.3)	496 (73.8)	26 (3.9)
3. Inborn metabolic diseases (n = 677) (false)	246 (36.3)	400 (59.1)	31 (4.6)
4. Diabetes (n = 683) (false)	46 (6.7)	357 (52.3)	280 (41)
5. Asthma (n = 675) (false)	70 (10.4)	394 (58.3)	211 (31.3)
6. Cancer (n = 667) (false)	94 (14.1)	458 (68.7)	115 (17.2)
7. NBS aims to identify a baby with a treatable disease before the onset of symptoms and assess treatment as soon as possible (n = 701) (true)	560 (79.9)	133 (19)	8 (1.1)
8. NBS is conducted using 4–6 drops of blood spots (n = 700) (true)	136 (19.4)	545 (77.9)	19 (2.7)
9. If the NBS result is positive, parents should promptly bring their babies to the hospital (n = 699) (true)	636 (91)	58 (8.3)	5 (0.7)
10. Positive NBS is a diagnostic test without a confirmatory test (n = 701) (false)	188 (26.8)	368 (52.5)	145 (20.7)
11. Babies with a screened disease might look healthy when they are born (n = 702) (true)	208 (29.6)	461 (65.7)	33 (4.7)
12. If the baby is not screened, the baby with the disease can have a severe intellectual disability or even die (n = 702) (true)	272 (38.8)	403 (57.4)	27 (3.8)

**Table 4 IJNS-09-00025-t004:** Comparison of participant characteristics between participants with good knowledge and those with poor knowledge.

	Knowledge		Univariate Analysis		Multivariate Analysis	
Factor	Good, n (%)	Poor, n (%)	OR (95% CI)	*p*-Value	OR (95% CI)	*p*-Value
	n = 74	n = 643				
**Gender**						
Female	58 (9.7)	539 (90.3)	1			
Male	16 (13.3)	104 (86.7)	1.43 (0.79–2.58)	0.237		
**Age, yr**						
≤20	3 (6.8)	41 (93.2)	1			
21–34	47 (10.7)	392 (89.3)	1.64 (0.49–5.50)	0.424		
≥35	24 (10.3)	210 (89.7)	1.56 (0.45–5.43)	0.483		
**Number of children**						
Pregnancy with first child	25 (11.7)	188 (88.3)	1.25 (0.75–2.10)	0.389		
Parents of one child or more	47 (9.6)	443 (90.4)	1			
**Place**						
Antenatal care clinic	36 (9.6)	339 (90.4)	1			
Postpartum ward	23 (10.2)	203 (89.8)	1.07 (0.61–1.85)	0.818		
Pediatric OPD and IPD	15 (16.5)	76 (83.5)	1.86 (0.97–3.57)	0.062		
**Education level**						
Below bachelor’s degree	25 (7.5)	307 (92.5)	1		1	
Bachelor’s degree	37 (11.9)	274 (88.1)	1.66 (0.97–2.83)	0.063	1.57 (0.92–2.70)	0.101
Master’s degree or higher	10 (17.2)	48 (82.8)	2.56 (1.16–5.66)	0.02	2.56 (1.14–5.75)	0.022 *
**Occupation**						
Healthcare provider	18 (25.4)	53 (74.7)	3.61 (1.98–6.59)	< 0.001	3.57 (1.94–6.56)	< 0.001 *
Non-healthcare provider	55 (8.6)	585 (91.4)	1		1	
**Income (THB)**						
<11,000	7 (7.1)	91 (92.9)	1			
11,000–24,999	25 (7.8)	294 (92.2)	1.11 (0.46–2.64)	0.821		
≥ 25,000	33 (16.6)	166 (83.4)	2.58 (1.10–6.08)	0.029		
**Previous child with a genetic disease**						
Yes	2 (9.5)	19 (90.5)	0.91 (0.21–3.99)	0.901		
No	72 (10.4)	623 (89.6)	1			

* significant difference *p*-value < 0.05.

**Table 5 IJNS-09-00025-t005:** Results of the parental attitudes toward the current and future NBS.

Questions/Statement	Strongly Disagree or Disagree n (%)	Neither Agree nor Disagree n (%)	Strongly Agree or Agree n (%)
**Attitude toward the current NBS**			
1. I still want my baby to have NBS despite the possibility of a false-positive result, which means that the screening result indicates a high risk for the disease but the baby does not have it (n = 709)	54 (7.6)	154 (21.7)	501 (70.7)
2. Even though no illness is confirmed after a positive NBS, I am still anxious, which may have an impact on my childcare (n = 704)	121 (17.2)	183 (26)	400 (56.8)
3. Only treatable conditions should be included in NBS (n = 701)	366 (52.2)	160 (22.8)	175 (25)
4. I would prefer to have physicians/nurses inform me before NBS test is conducted (n = 706)	12 (1.7)	29 (4.1)	665 (94.2)
5. Parental consent is necessary before blood sampling for NBS (n = 704)	18 (2.5)	45 (6.4)	641 (91.1)
6. I permit the use of self-care NBS specimens for future research without personal identification (n = 709)	100 (14.1)	188 (26.5)	421 (59.4)
** Attitudes toward future NBS **			
7. When it is available, I want my baby screened for an additional 30–40 treatable inborn metabolic disorders (so-called expanded NBS) (n = 703)	8 (1.1)	92 (13.1)	603 (85.8)
8. In the event that expanded NBS may indicate that I have a previously undiagnosed health condition, I still want NBS to be conducted, and I also want to be informed about my discovered health condition (n = 703)	9 (1.3)	74 (10.5)	620 (88.2)
9. The government should cover expanded NBS for all newborn babies (n = 703)	12 (1.7)	76 (10.8)	615 (87.5)
10. For the expanded NBS for my baby, I am willing to pay out of pocket if necessary (n = 686)	69 (10.1)	168 (24.5)	449 (65.4)
11. If screening for incurable diseases is available (and included) in the NBS panel, I would like to have my baby screened for those conditions (n = 689)	26 (3.8)	93 (13.5)	570 (82.7)
12. If screening for adult-onset diseases is available (and included) in the NBS panel, I would like to have my baby screened for those too (n = 673)	17 (2.5)	91 (13.5)	565 (84)

## Data Availability

De-identified collected data and the final Thai questionnaire in this study can be shared upon reasonable request.

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
