# Peer review of "Parental Awareness, Knowledge, and Attitudes Regarding Current and Future Newborn Bloodspot Screening: The First Report from Thailand"

_2409-515X, 2023, doi:10.3390/ijns9020025_

Round 1

Reviewer 1 Report

The authors propose an evaluation of awareness, knowledge, and attitude of the parents regarding the newborn screening. The paper is easy to read and understand. It allows an evaluation of the patients' knowledge and information needs. 

The authors make a distinction between awareness and knowledge which is not very clear and for which I do not see the need to make a distinction. Awareness would only be for very general information, whereas knowledge would be for more specific information. Therefore, putting a specific table on awareness seems superfluous, leading to the belief that this information is as important as knowledge. It might be interesting to note in the text the results on awareness - i.e., the general knowledge of the parents - but leave out table 2 and keep the detail for specific knowledge. 

Some references could be added/updated in relation to newborn screening around the world. 

Finally, the discussion could be enriched with an ethical perspective, especially on the best interest of the child. The scientific community could benefit from this reflection on the appropriateness of offering opt-in or opt-out screening, especially when we see the extremely low level of knowledge of parents.

Minor remarks: 

- The term NBS usually refers to "Newborn screening" and not to "newborn bloodspot screening  

- Line 39: NBS is dedicated to treatable diseases and not manageable conditions, it is possible to use Junger and Wilson criteria if needed. It can be added that in some countries, the criteria can be broader and add "manageable" conditions as in some states in the USA.

- L40: The reference on the different screened conditions can be updated (more recent article by Loeber: Loeber, J.G. et al. (2021) Neonatal Screening in Europe Revisited: An ISNS Perspective on the Current State and Developments Since 2010. Int. J. neonatal Screen. 7)

- L42: 60 years since the start of NBS

- L97: incomplete sentence

- Table 1: Add meaning of abbreviations (ANC)

 - L 120: Add correspondence between Bath and $ on first occurrence

- Table 4: typo: "Baht

 - If table 2 and 4 maintained: present the items in the same order.

 - The Czech study to which you refer (Frankova et al), if it is interested in "awareness", is in fact the evaluation of knowledge which you include in the 2 categories: awareness and knowledge. The comparison of the 2 populations (L194-196) is therefore inappropriate. 

- L204: “Similar” is not appropriate, as it is not at the same time that parents wish to receive information.

- L230-250: The discussion about giving parents the choice by opt-in when they are not aware of it seems to be demagogic. Wouldn't it be better to consider the primary interest of the child? while authorizing the opt-out? 

It is important to differentiate between the extension of screening, as has been done in Thailand, to treatable diseases, for which the opt-out seems the best option given the lack of knowledge of the general population, and an extension to untreatable diseases, even those that appear late in life. 

Is this screening, even if the parents wish it, really beneficial to the child and is it practically possible, are genetic consultations really able to receive and orient parents for hypothetical future diseases or towards late palliative care?

Furthermore, it would be interesting to know what are the modalities of extension of the current NBS: with or without consent? and with which information is put in place?

Reviewer 2 Report

The authors of the manuscript ‘Parental Awareness, Knowledge, and Attitude Regarding the Current and Future Newborn Bloodspot Screening: the First Report from Thailand’ provide an important overview of the opinion of parents on NBS supported by a clear study/questionnaire design. The paper is highly relevant for the screening community as very limited is available on the parental perspectives in Asia.. The manuscript is well-written and presentation of the work is of quality. My compliments to the authors for their important work.

Methods

The method section is well written, the development of the questionnaire is thorough and definitions of 'good awereness/good knowlegde' are well defined.

Results

Can the authors elaborate on their choice to define educational level? In many questionnaire studies a Bachelor level would already be higher vocational education excluding the low/middle groups. Low being primary education, lower vocational education, lower and middle general secondary education. Middle being middle vocational education, higher secondary education, and pre-university education. Choosing ‘below Bachelor level’ might not be a true representation of this group and your statements that educational level is not a significant factor.

Line 167. Consent. How is this currently done in Thailand? Do parents have to provide written consent for NBS? What is your participation rate? Could you add this to the introduction?

169. Left over blood. How is this currently done in Thailand?

Table 5. Did you provide parents with some additional background information before these questions? It seems that some of these questions might be quite difficult (e.g. what is false-positive results) with only 10% of participants with good knowledge scores?

Line 172: About 57% of the parents still have anxiety which might affect 172 their care if their children have a false positive for NBS. This is a very hypothetical question. You are asking parents if in the hypothetical situation that their newborns has a false-positive screening result, if they would experience anxiety. Do you think these outcomes are representable if knowledge scores about NBS are relatively low?

Discussion

Line 199. The authors make a comparison to the study in the Philipines, however this study uses a very different categorization of education level. Outcomes can therefore not be compared.

The reference list is somewhat limited, there a are a lot of other studies who have looked into parental perspectives (with regard to SCID, CF, expanded NBS with NGS). I would recommend including more references and compare data to other studies as well in the discussion.

Minor

- Line 97: comma in middle of sentence

Reviewer 3 Report

The manuscript by Wilaiwongsathien, et al provides a novel look into parental awareness and attitudes around NBS in Thailand. Overall, the manuscript is well-written and presents an important insight into family perspectives on NBS.

The following suggestions are recommended to enhance the paper:

1) Introduction: It would be helpful for the authors to expand the introduction to discuss why family awareness may be low and why it is important for families to understand and have awareness of NBS. Likewise, it would be helpful to know some characteristics about the Thai program (is consent needed? Are DBS stored? Etc?)

2) Line 97-98 reads oddly. Please reword for clarity/remove typos.

3) Line 100-102 needs clarification. Perhaps indicate that the questions covered topics pertinent to the current NBS program and topics that may be considered for future aspects in NBS.

4) Results. It would be helpful to know how the demographics of the participants correspond with demographics more generally in Thailand.

5) Lines 150-151 needs clarification. I think it is meant to say that those diseases can't screened for by NBS

6) Table 3: CH is misspelled: Congenital Hypothyroidism

7) Lines 169-173: Can use some clarification - are DBS used for research? Why do 57% have anxiety - from the false positive result? This is not clear.

8) Discussion: A bit hard to follow and could use some improved grammar.

9) Lines 202-207: These contradict each other. The study presented in the paper had more people indicate that the 1st trimester was preferred, but the references suggest the 3rd trimester was preferred.
